# Diarrhea in Yemeni children under five: A multi-level analysis of population-based surveys, 1991–2022

Zahir M. Tag[1*], Laith J. Abu-Raddad[1,2,3,4], Hiam Chemaitelly[1,2*]

1 Infectious Disease Epidemiology Group, Weill Cornell Medicine-Qatar, Cornell University, Doha, Qatar, 2 Department of Population Health Sciences, Weill Cornell Medicine, Cornell University, New York, New York, United States of America, 3 Department of Public Health, College of Health Sciences, QU Health, Qatar University, Doha, Qatar, 4 College of Health and Life Sciences, Hamad bin Khalifa University, Doha, Qatar

* zat4002@qatar-med.cornell.edu (ZMT); hsc2001@qatar-med.cornell.edu (HC)

## Abstract

### Background

Yemen, grappling with a protracted humanitarian crisis, armed conflict, forced displacements, and economic hardship, faces a burden of childhood diarrhea. This study aimed to assess diarrhea prevalence, trends, and underlying factors among children under five in this population.

### Methods

Data were sourced from the population-based, nationally representative Yemen Demographic and Health Surveys (1991 and 2013) and Multiple Indicator Cluster Surveys (2006 and 2022). A three-level random-effects logistic regression model was used to identify risk factors, with clustering effects analyzed at both the neighborhood and household levels.

### Results

The study included 6,451 children under five in 1991, 3,778 in 2006, 15,278 in 2013, and 19,501 in 2022. Diarrhea prevalence was 34.8% (95% CI: 32.6–37.1%) in 1991, 33.6% (95% CI: 31.6–35.6%) in 2006, 31.4% (95% CI: 30.1–32.7%) in 2013, and 37.4% (95% CI: 36.2–38.7%) in 2022. The adjusted odds ratios (aORs) for diarrhea were twice as high for children aged 12–23 months compared to those aged less than 12 months and decreased steadily with increasing age. Females had 15% lower odds of diarrhea than males. The aORs were higher for households in North Yemen, those without water on premises, or those with unimproved toilet facilities, but lower for households with more than five members and those in the highest wealth quintile. Between-cluster differences decreased between 1991 and 2006 but increased

**Data availability statement:** Data are publicly available for research purposes through the DHS program and UNICEF MICS at https://dhsprogram.com/ and https://mics.unicef.org/, respectively. DHS data can be requested at https://dhsprogram.com/data/new-user-registration.cfm, and MICS data at https://mics.unicef.org/surveys, with free registration and approval required.

**Funding:** This work was supported by the Specialized Biomedical Research Training Program, the Biomedical Research Program, and the Biostatistics, Epidemiology, and Biomathematics Research Core, all at Weill Cornell Medicine-Qatar, with funding provided to ZMT, LJA-R, and HC. The funders had no role in study design, data collection and analysis, decision to publish, or preparation of the manuscript.

**Competing interests:** The authors have declared that no competing interests exist.

between 2013 and 2022. Disparities were much more pronounced between households than across neighborhoods.

## Conclusions

Using a three-level modeling approach and analyzing trends over a three-decade period, this study revealed a persistent and worsening burden of childhood diarrhea in Yemen, with prevalence more than twice the global average. Urgent action is needed to improve water and sanitation infrastructure and implement targeted programs to reduce diarrhea prevalence.

## Author summary

In Yemen, a country challenged by an ongoing humanitarian crisis, armed conflict, and economic hardship, childhood diarrhea remains a serious public health issue. This study investigated diarrhea prevalence, trends, and underlying factors among Yemeni children under five. It analyzed data from four national surveys conducted in 1991, 2006, 2013, and 2022. The findings revealed that over one-third of children were affected by diarrhea, with the situation worsening over time. Key risk factors identified were the child's age, residence in North Yemen, lack of access to water on premises, unimproved toilet facilities, smaller household size, and low wealth quintile. Disparities in diarrhea prevalence increased over the last decade and were more pronounced between households than across neighborhoods. The latter highlights the pivotal role of household-level socioeconomic factors and living conditions in determining diarrhea risk. These findings underscore the urgent need to improve water and sanitation infrastructure and implement tailored programs to reduce the burden of diarrhea across Yemen.

## Introduction

Diarrhea, primarily caused by pathogens in contaminated food and water and defined as at least three loose or watery bowel movements in a day [1–3], is the third leading cause of death in children under five, claiming 445,000 lives annually worldwide [3]. This burden disproportionately affects low- and middle-income countries, where 90% of these deaths occur [3,4].

Despite efforts to achieve Sustainable Development Goal (SDG) 6, which aims to "ensure availability and sustainable management of water and sanitation for all" [5], gaps persist. As of 2022, 2.2 billion people lacked access to safe drinking water and basic handwashing facilities, while 3.5 billion lacked safely managed sanitation [5]. This shortfall hampers efforts to reduce diarrhea disease burden, thus impeding progress toward the SDG 3 target of "ending preventable deaths of newborns and children under five by 2030" [6].

These structural and environmental factors are compounded by various demographic, socio-economic, and behavioral factors [7]. As identified in a systematic review [7], these encompass factors such as the mother's age, education, and occupation, the child's age and sex, household income, living conditions such as crowding, and practices such as feeding methods and handwashing habits.

Located on the southern part of the Arabian Peninsula, Yemen, with a population of 39.4 million [8], is one of the most impoverished countries in the Middle East [9]. As of 2023, 40% of its population was under 15 years of age and 16% was under five [8]. Until 2013, Yemen made steady progress in maternal and child health, control of communicable diseases such as malaria, and overall life expectancy [10]. Notably, the child mortality rate declined from 58 per 1,000 live births in 1990–15 in 2016 [10]. However, neonatal complications, lower respiratory tract infections, and diarrheal diseases remained leading causes of child mortality and years of life lost [11].

Since 2015, Yemen's ongoing conflict has dramatically exacerbated the country's long-standing economic and health challenges. From 2015 to 2022, the gross domestic product per capita decreased by 52%, leaving two-thirds of the population needing humanitarian aid [9]. Approximately 18 million Yemenis lack access to safe drinking water and adequate sanitation facilities [9], fueling multiple outbreaks of acute diarrhea and cholera since 2016 [12].

Despite the disease burden, the determinants of diarrheal morbidity in children under five in Yemen remain inadequately understood. To date, only two studies, to our knowledge, have investigated risk factors for childhood diarrhea in the country, both limited in scope to specific hospitals or regions [13,14]. Although studies in other conflict-affected countries have used nationally representative surveys, these have typically relied on cross-sectional data from single time points [15–17].

By leveraging three decades of repeated, nationally representative population-based surveys, this study aims to fill this gap through an in-depth investigation of individual- and household-level factors contributing to diarrhea in children under five, along with assessments of regional and household disparities and of temporal trends.

## Methods

### Data source and study population

Data for this study were sourced from Yemen's 1991 and 2013 Demographic and Health Surveys (DHS) [18]. These surveys were implemented by Yemen's Central Statistical Organization (CSO) and the Ministry of Public Health and Population (MOPHP), with technical and financial support from the DHS program, which operates under the United States Agency for International Development (USAID). Additional data were sourced from the 2006 and 2022 Multiple Indicator Cluster Surveys (MICS) [19], implemented by the same national institutions with support from the United Nations Children's Fund (UNICEF). Both DHS and MICS are nationally representative population-based cross-sectional surveys that employ comparable sampling and data collection methodologies [20,21].

Efforts were made to obtain the 1997 DHS dataset; however, despite multiple requests and follow-ups submitted to the CSO and MOPHP, the data could not be accessed. The lack of response may reflect the ongoing conflict in Yemen, which continues to limit institutional capacity and hinder collaboration on research efforts.

Participants were selected through a two-stage cluster sampling design [18,19]. In the first stage, primary sampling units (PSUs) from each governorate were randomly selected based on probability proportional to size from enumeration areas used in the most recent national census. To reduce underrepresentation, PSUs with smaller populations and/or located in rural areas were oversampled. In the second stage, approximately 20–30 households from each PSU were selected through systematic random sampling. All women aged 15–49 years within identified households were eligible for participation.

The collected data encompassed household characteristics, living conditions, socio-demographic information, data on maternal, child, and reproductive health, as well as fertility and mortality indicators. Unlike DHS, which included only children under five living with biological parents, MICS considered all household children aged up to 17 years, including

orphans and foster children. For this study, the population consisted of children under five years of age from the available DHS and MICS surveys. Household-level and individual-level data were retrieved by merging household, women, and children data files following DHS and MICS statistical guidelines [22,23].

## Primary outcome

The primary outcome was the occurrence of diarrhea in children under five within the two weeks preceding the survey. Mothers or caregivers were asked, "Has (name of child) had diarrhea in the last two weeks?" Responses were recoded into a binary variable: 1 for "Yes" and 0 for "No". Indeterminate responses such as "Missing" and "Don't know", were excluded from the analysis.

## Individual- and household-level factors

Individual- and household-level factors potentially associated with diarrhea were determined *a priori*, informed by the existing literature [24–32]. Individual-level factors included the child's sex, age in months, and maternal education (none, basic, intermediate-advanced).

Household-level factors included the type of place of residence (rural, urban), region (North, South), number of individuals in household, wealth index (lowest to highest quintiles), cooking place (house, separate building, other), and water, sanitation and hygiene (WASH) factors [33,34] such as source of drinking water (unimproved, improved), time to water source (on premises, ≤ 30 minutes, > 30 minutes), type of toilet facility (unimproved, improved), and water treatment (yes, no).

Unimproved water sources comprised regular wells, unprotected water surfaces, rivers, tanker trucks, and containerized water, whereas improved sources comprised government and local network supplies, tube and pumped wells, rainwater, and bottled water. Unimproved toilet facilities encompassed pit latrines, flush toilets without sewer connections, open-drain toilets, street toilets, bucket toilets, shared facilities, and open defecation, while improved facilities encompassed flush toilets with piped sewer connections or septic tank systems.

Since a wealth index was not available in the 1991 DHS, the measure was constructed using factor analysis as per DHS guidelines [35,36]. The analysis was performed using the variable list employed by the DHS to construct the wealth index for the 2013 survey round. These included household assets and amenities such as air conditioning, bicycle, blender, car, color television, dwelling type, electric fan, electricity, flooring material, gas or electric stove, motorcycle, number of sleeping rooms, primary source of drinking water, radio, refrigerator, sewing machine, telephone, television, type of toilet facility, vacuum cleaner, video player, washing machine, and water heater.

## Statistical analysis

**Estimation of diarrhea prevalence.** The individual and household characteristics of children included in the study were described using frequency distributions. DHS and MICS sample probability weights were applied in estimating the prevalence of diarrhea, following DHS [22] and MICS [37] statistical guidelines, to correct for unequal participant selection and ensure the sample's representativeness of the broader population.

**Model selection and goodness of fit.** Associations with diarrhea were investigated using chi-square tests and univariable logistic regression analyses, factoring the complex survey design [38]. Variables with a p-value ≤0.2 in the univariable analyses were considered for inclusion in the multivariable model. A three-level random-effects logistic regression model was then constructed to account for the hierarchical structure of the data, including clustering and the non-independence of observations among children within households and communities [39]. Individual-level probability weights from the DHS and MICS surveys were applied at the first level.

Model construction followed established multilevel modeling approaches employed in prior research [24–27,40,41]. Model selection involved comparing the goodness-of-fit across five models using Akaike's Information Criterion (AIC) and Bayesian Information Criterion (BIC), with lower values indicating better fit [42].

The first four models were two-level that included, respectively, intercept only, individual-level factors only, household-level factors only, and both individual- and household-level factors at the first level, and PSUs at the second level. The fifth model expanded to three-levels, with individual- and household-level factors included at the first level, household units at the second level, and PSUs at the third level. Multicollinearity was assessed using the variance inflation factor (VIF), with values >5 indicating multicollinearity [43]. Interactions were not investigated.

**Multilevel logistic regression analyses.** A p-value of <0.05 in the multivariable analysis indicated a statistically significant association with diarrhea. Odds ratios (ORs), adjusted ORs (aORs), and corresponding 95% confidence intervals (CIs) were reported.

The intraclass correlation coefficient (ICC) was calculated to assess the proportion of the total variance attributable to the clustering effects of each of household units and PSUs, with ICC ≥ 0.02 indicating potential clustering effects [44].

The median odds ratio (MOR) was also calculated to quantify the median value of the distribution of odds ratios, comparing the odds of diarrhea between individuals in two randomly selected clusters—one with high risk and the other with low risk [45].

**Sensitivity analysis.** Unlike in the DHS, some factors, such as birthweight (≤2, > 2 kilograms), breastfeeding status (yes, no), and health card possession (yes, no) were only collected for children under three in MICS. Given potential associations with diarrhea [46,47], a sensitivity analysis was conducted by replicating the main analysis while incorporating these additional factors, but considering only children under three across the different surveys.

**Missing data handling.** Missing values for all selected variables across the four surveys were at or below 5%, which is generally considered an acceptable threshold for complete case analysis [48,49]. Therefore, no imputation was performed.

**Meta-analysis of diarrhea prevalence in other countries with available DHS.** To compare estimates of diarrhea prevalence in Yemen with those from other countries with available DHS, 276 prevalence measures from 82 countries spanning 1985–2024 were obtained through DHS STATcompiler [50], a public database-building tool enabling cross-country and temporal comparisons of DHS indicators. Although the complete dataset for Yemen's 1997 DHS was restricted, the diarrhea prevalence estimate was available and included in the comparative analysis.

Meta-analyses were conducted to estimate the pooled mean diarrhea prevalence for each World Health Organization (WHO) region [African (AFR), Americas (AMR), Eastern Mediterranean (EMR), European (EUR), South-East Asia (SEAR), and Western Pacific (WPR)] by 10-year intervals, and globally by 5-year intervals. For countries with multiple DHS rounds within a given interval, all eligible estimates were included to capture temporal variability and improve the precision of regional and global pooled estimates.

To address variance instability, particularly for prevalence estimates near 0 or 1, the Freeman–Tukey double arcsine square-root transformation was applied—a method widely used in meta-analyses of proportions [51]. This approach was selected based on its demonstrated methodological adequacy for stabilizing variance [52].

Transformed prevalences were then weighted using the inverse variance method [53] and pooled using DerSimonian–Laird random-effects models to account for between-study heterogeneity [54]. Heterogeneity was assessed using $I^2$, which quantifies the proportion of variation attributable to true differences in prevalence measures across surveys rather than chance [55].

**Statistical software.** Descriptive and regression analyses were performed using Stata/SE version 18.0 (Stata Corporation, College Station, TX, USA). Meta-analyses were performed using R 4.3.1 meta package.

## Ethical considerations

Ethical approval was not required to conduct this study, as both DHS and MICS are de-identified, publicly available datasets.

## Results

### Study population characteristics

The study population comprised 6,451 children under five in the 1991 DHS, 3,778 in the 2006 MICS, 15,278 in the 2013 DHS, and 19,501 in the 2022 MICS (Table 1). In all surveys, children were almost evenly distributed by sex and across the first 5 years of life.

The majority of children lived in North Yemen, and over 70% resided in rural areas. Maternal education increased from 9.2% in 1991 to 59.5% in 2022. Access to improved drinking water reached 79.2% in 2022, up from an average of 56% in earlier surveys. Access to water on premises doubled over time, increasing from 31.8% in 1991 to 60.0% in 2022. Two-thirds of the population had access to improved toilet facilities in 2022 compared to <10% in 1991.

### Diarrhea prevalence

Diarrhea prevalence among children under five was 34.8% (95% CI: 32.6-37.1%) in 1991, 33.6% (95% CI: 31.6-35.6%) in 2006, 31.4% (95% CI: 30.1-32.7%) in 2013, and reached 37.4% (95% CI: 36.2-38.7%) in 2022 (Fig 1).

Prevalence varied across age groups, peaking among children aged 12–23 months and decreasing with older age (Table 1). Geographical differences were noted, with diarrhea being consistently more prevalent in rural areas and North Yemen. Conversely, prevalence was lowest among children in the highest wealth quintile and those with access to improved drinking water, water on premises, and improved toilet facilities.

### Model selection and goodness of fit

For each survey round, factors associated with diarrhea in univariable analyses were included in the multivariable analysis (Tables 2 and 3). The multivariable model structure was determined by comparing the goodness-of-fit across five models.

S2–S5 Tables present the results of these models for the different survey rounds. Two-level models incorporating either individual-level factors alone, household-level factors alone, or both at the first level, along with PSUs at the second level, showed only slight improvements in model fit, as indicated by the reductions in AIC and BIC.

Meanwhile, the three-level model, including both individual- and household-level factors at the first level, household units at the second level, and PSUs at the third level, yielded the best fit for all survey rounds, with substantial reductions in AIC and BIC (S2–S5 Tables). VIF was < 5 across all models, indicating no multicollinearity among predictor variables.

### Associations with recent diarrhea

Tables 2 and 3 include the results of the multilevel multivariable logistic regression analyses.

*Individual-level factors.* The aORs of diarrhea were highest for children aged 12–23 months compared to those aged <12 months across all surveys and declined steadily with increasing age. In 2022, for example, the aOR was 1.73 (95% CI: 1.42-2.11) for ages 12–23 months, 0.82 (95% CI: 0.68-0.98) for 24–35 months, 0.55 (95% CI: 0.45-0.68) for 36–47 months, and 0.38 (95% CI: 0.31-0.46) for 48–59 months.

In recent surveys, females had approximately 15% lower odds of experiencing diarrhea compared to males. The association between maternal education and recent diarrhea did not follow a consistent pattern and often lacked statistical significance.

*Household-level factors.* Children living in North Yemen consistently had significantly higher odds of diarrhea compared to those living in South Yemen, with aOR of 11.10 (95% CI: 6.74-18.27) in 1991, 1.98 (95% CI: 1.31-3.00) in 2006, 2.31 (95% CI: 1.86-2.88) in 2013, and 4.60 (95% CI: 3.61-5.86) in 2022. Meanwhile, no consistent pattern was observed for the odds of diarrhea between rural and urban settings.

Larger households including >5 individuals were generally associated with lower odds of diarrhea compared to smaller ones, although this association did not always reach statistical significance. Children in the highest wealth quintile exhibited lower odds of diarrhea than those in the lowest wealth quintile.

# Table 1. Characteristics of study participants in children under 5 years of age in Yemen DHS and MICS.

| Year | DHS 1991 | | | MICS 2006 | | | DHS 2013 | | | MICS 2022 | | |
|---|---|---|---|---|---|---|---|---|---|---|---|---|
| Characteristics | Total | Recent diarrhea | | Total | Recent diarrhea | | Total | Recent diarrhea | | Total | Recent diarrhea | |
| | N (%*) | N (%*) | χ² p-value | N (%*) | N (%*) | χ² p-value | N (%*) | N (%*) | χ² p-value | N (%*) | N (%*) | χ² p-value |
| **I. Individual-level factors** | | | | | | | | | | | | |
| Sex of child | | | | | | | | | | | | |
| Male | 3,308 (51.4) | 1,077 (35.8) | 0.004 | 1,927 (50.9) | 686 (34.7) | 0.146 | 7,827 (51.2) | 2,544 (32.6) | 0.009 | 10,071 (51.5) | 3,536 (38.7) | 0.004 |
| Female | 3,143 (48.6) | 957 (33.7) | | 1,851 (49.1) | 615 (32.4) | | 7,451 (48.8) | 2,226 (30.2) | | 9,430 (48.5) | 3,161 (36.1) | |
| Current age of child (Months) | | | | | | | | | | | | |
| 0-11 | 1,483 (23.2) | 503 (36.8) | <0.001 | 850 (22.6) | 315 (36.1) | <0.001 | 3,226 (21.2) | 1,146 (35.9) | <0.001 | 3,899 (20.3) | 1,433 (40.7) | <0.001 |
| 12-23 | 1,223 (19.0) | 497 (44.2) | | 715 (19.1) | 320 (43.4) | | 3,044 (20.1) | 1,395 (46.2) | | 3,734 (20.1) | 1,662 (48.7) | |
| 24-35 | 1,388 (21.3) | 471 (37.4) | | 749 (19.6) | 277 (36.6) | | 3,066 (20.2) | 984 (31.8) | | 4,020 (21.0) | 1,391 (36.7) | |
| 36-47 | 1,264 (19.7) | 319 (28.2) | | 781 (20.5) | 226 (28.0) | | 2,978 (19.0) | 727 (24.7) | | 4,077 (20.5) | 1,208 (32.0) | |
| 48-59 | 1,093 (16.8) | 244 (25.6) | | 683 (18.2) | 163 (23.2) | | 2,940 (19.5) | 510 (17.2) | | 3,729 (18.1) | 996 (28.2) | |
| Mother's education | | | | | | | | | | | | |
| None | 5,603 (90.8) | 1,877 (36.2) | <0.001 | 2,499 (66.8) | 866 (33.6) | 0.990 | 8,502 (55.1) | 2,668 (31.5) | 0.235 | 8,430 (40.5) | 3,015 (38.5) | 0.027 |
| Basic | 500 (5.9) | 106 (22.3) | | 968 (24.8) | 328 (33.4) | | 1,294 (8.3) | 386 (28.5) | | 4,758 (25.5) | 1,687 (38.6) | |
| Intermediate-Advanced | 342 (3.3) | 48 (17.4) | | 310 (8.4) | 106 (33.6) | | 5,482 (36.6) | 1,716 (32.0) | | 6,307 (34.0) | 1,995 (35.3) | |
| **II. Household-level factors** | | | | | | | | | | | | |
| Place of residence | | | | | | | | | | | | |
| Urban | 1,433 (16.5) | 332 (26.2) | <0.001 | 954 (27.0) | 280 (29.2) | 0.007 | 3,534 (27.2) | 977 (28.3) | 0.014 | 4,620 (27.5) | 1,267 (32.6) | <0.001 |
| Rural | 5,018 (83.5) | 1,702 (36.5) | | 2,824 (73.0) | 1,021 (35.2) | | 11,744 (72.8) | 3,793 (32.6) | | 14,881 (72.5) | 5,430 (39.3) | |
| Region | | | | | | | | | | | | |
| South Yemen | 1,559 (13.8) | 189 (12.1) | <0.001 | 579 (14.3) | 152 (24.7) | <0.001 | 3,336 (14.0) | 748 (20.8) | <0.001 | 5,119 (14.4) | 694 (16.8) | <0.001 |
| North Yemen | 4,892 (86.2) | 1,845 (38.4) | | 3,199 (85.7) | 1,149 (35.1) | | 11,942 (86.0) | 4,022 (33.1) | | 14,382 (85.6) | 6,003 (40.9) | |
| No. of individuals in household | | | | | | | | | | | | |
| 1-5 | 1,238 (19.4) | 405 (36.2) | 0.427 | 751 (19.9) | 292 (37.7) | 0.101 | 3,647 (25.8) | 1,228 (33.4) | 0.159 | 4,927 (27.4) | 1,739 (38.1) | <0.001 |
| 6-10 | 3,650 (57.2) | 1,170 (35.1) | | 1,918 (51.0) | 651 (33.0) | | 7,443 (49.7) | 2,260 (30.5) | | 9,872 (50.4) | 3,590 (38.9) | |
| 11-15 | 1,203 (17.9) | 361 (33.4) | | 753 (19.3) | 241 (31.7) | | 2,838 (17.7) | 868 (31.2) | | 3,230 (16.0) | 995 (34.9) | |
| >15 | 360 (5.5) | 98 (30.7) | | 356 (9.8) | 117 (32.2) | | 1,350 (6.8) | 414 (31.0) | | 1,472 (6.2) | 373 (29.0) | |

*(Continued)*

| Year | DHS 1991 | | | MICS 2006 | | | DHS 2013 | | | MICS 2022 | | |
|---|---|---|---|---|---|---|---|---|---|---|---|---|
| Characteristics | Total | Recent diarrhea | | Total | Recent diarrhea | | Total | Recent diarrhea | | Total | Recent diarrhea | |
| | N (%*) | N (%*) | χ² p-value | N (%*) | N (%*) | χ² p-value | N (%*) | N (%*) | χ² p-value | N (%*) | N (%*) | χ² p-value |
| Wealth index† | | | | | | | | | | | | |
| Lowest | 1,214 (20.9) | 474 (41.4) | <0.001 | 809 (23.2) | 296 (35.1) | 0.019 | 3,334 (22.5) | 1,106 (32.6) | <0.001 | 4,981 (23.7) | 2,343 (44.7) | <0.001 |
| Second | 1,119 (19.2) | 408 (38.2) | | 798 (21.4) | 300 (36.3) | | 3,393 (21.4) | 1,123 (34.0) | | 3,766 (20.5) | 1,516 (43.0) | |
| Middle | 1,317 (22.0) | 447 (35.9) | | 829 (20.1) | 298 (35.2) | | 3,266 (20.2) | 1,103 (33.3) | | 3,694 (19.2) | 1,209 (37.3) | |
| Fourth | 1,237 (19.4) | 376 (33.7) | | 767 (19.5) | 251 (32.7) | | 3,005 (18.6) | 879 (30.5) | | 3,643 (19.5) | 985 (34.9) | |
| Highest | 1,564 (18.5) | 329 (23.5) | | 575 (15.8) | 156 (26.6) | | 2,280 (17.3) | 559 (25.3) | | 3,375 (17.1) | 637 (23.7) | |
| Cooking place | | | | | | | | | | | | |
| House | 4,215 (64.4) | 1,260 (33.0) | 0.005 | 2,319 (62.0) | 788 (33.3) | 0.886 | 10,530 (69.2) | 3,219 (31.3) | 0.954 | 11,300 (53.7) | 3,287 (33.8) | <0.001 |
| Separate building | 1,944 (31.2) | 693 (39.2) | | 1,430 (37.7) | 503 (34.2) | | 3,578 (21.2) | 1,138 (31.6) | | 8,128 (46.0) | 3,383 (41.7) | |
| Other | 273 (4.4) | 78 (30.9) | | 14 (0.3) | 6 (36.1) | | 1,097 (9.6) | 388 (31.7) | | 40 (0.3) | 17 (37.1) | |
| *WASH-related factors* | | | | | | | | | | | | |
| Source of drinking water‡ | | | | | | | | | | | | |
| Unimproved | 2,623 (42.6) | 868 (36.7) | 0.126 | 1,626 (43.5) | 585 (35.2) | 0.181 | 7,745 (44.5) | 2,582 (34.5) | <0.001 | 4,746 (20.8) | 2,074 (43.2) | <0.001 |
| Improved | 3,828 (57.4) | 1,166 (33.3) | | 2,152 (56.5) | 716 (32.4) | | 7,518 (55.5) | 2,184 (28.9) | | 14,755 (79.2) | 4,623 (35.9) | |
| Time to water source (Minutes) | | | | | | | | | | | | |
| On premises | 2,225 (31.8) | 578 (29.9) | 0.001 | 1,868 (49.1) | 621 (32.7) | 0.627 | 12,641 (86.6) | 3,761 (30.0) | <0.001 | 11,177 (60.0) | 3,091 (33.0) | <0.001 |
| ≤30 | 2,123 (37.0) | 682 (34.0) | | 858 (21.7) | 306 (35.3) | | 578 (3.6) | 212 (39.5) | | 3,936 (20.6) | 1,688 (44.2) | |
| >30 | 1,814 (31.2) | 664 (40.0) | | 1,002 (29.2) | 355 (33.6) | | 1,623 (9.8) | 630 (38.6) | | 4,090 (19.4) | 1,832 (43.9) | |
| Type of toilet facility§ | | | | | | | | | | | | |
| Unimproved | 5,837 (93.3) | 1,928 (35.8) | <0.001 | 1,201 (34.5) | 432 (34.8) | 0.352 | 8,228 (53.2) | 2,733 (32.7) | 0.028 | 8,908 (40.0) | 3,663 (43.1) | <0.001 |
| Improved | 614 (6.7) | 106 (21.0) | | 2,576 (65.5) | 869 (32.9) | | 7,015 (46.8) | 2,028 (30.0) | | 10,591 (60.0) | 3,033 (33.7) | |
| Water treatment | | | | | | | | | | | | |
| No | 6,026 (94.6) | 1,904 (34.8) | 0.861 | 3,495 (92.6) | 1,210 (33.9) | 0.299 | 13,575 (90.3) | 4,183 (31.2) | 0.217 | 17,356 (90.6) | 5,881 (37.3) | 0.296 |
| Yes | 382 (5.4) | 116 (34.2) | | 261 (7.4) | 85 (30.0) | | 1,681 (9.7) | 581 (33.2) | | 2,092 (9.4) | 800 (39.1) | |
| Total (%) | 6,451 (100.0) | 2,034 (34.8) | NA | 3,778 (100.0) | 1,301 (33.6) (31.2) | NA | 15,278 (100.0) | 4,770 (31.4) | NA | 19,501 (100.0) | 6,697 (37.4) | NA |

*(Continued)*

**Table 1.** (Continued)

Abbreviations: DHS, demographic and health survey; MICS, multiple indicator cluster survey; NA, not applicable; No., number; WASH, Water/Sanitation/Hygiene.

\* Percentages were calculated applying DHS or MICS sampling weights to account for the complex survey design of the DHS or MICS study. Missing values were excluded from the analysis.

† The wealth index for the 1991 survey was constructed following DHS guidelines, employing factor analysis. The selection of variables for this analysis was informed by those employed by the DHS in constructing the wealth index for the 2013 Yemen DHS. These variables encompassed household assets and amenities such as air conditioning, bicycle, blender, car, color television, dwelling type, electric fan, electricity, flooring material, gas or electric stove, motorcycle, number of sleeping rooms, primary source of drinking water, radio, refrigerator, sewing machine, telephone, television, type of toilet facility, vacuum cleaner, video player, washing machine, and water heater.

‡ In classifying water sources, "unimproved" refers to sources such as regular wells, unprotected water surfaces, rivers, tanker trucks, and containerized water, while "improved" refers to government and local network supplies, tube and pumped wells, rainwater, and bottled water.

§ In classifying toilet facilities, "unimproved" refers to pit latrines, flush toilets without sewer connections, open-drain toilets, street toilets, bucket toilets, shared facilities, and open defecation, while "improved" refers to flush toilets with piped sewer connections or septic tank systems.

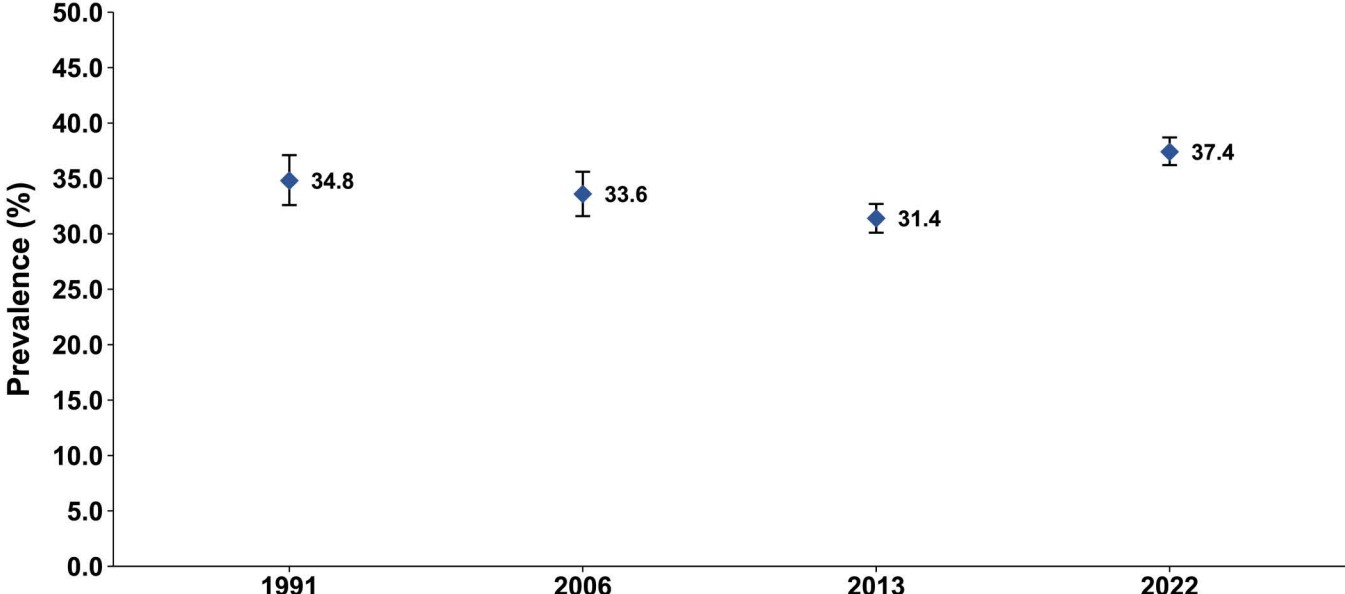

**Fig 1. Prevalence of recent diarrhea in children under 5 years of age in Yemen DHS and MICS.** Abbreviations: DHS, demographic and health survey; MICS, multiple indicator and cluster survey.

Recent surveys from 2013 and 2022 indicated that children living in households without water on premises had higher odds of diarrhea compared to those with water access on premises. In the most recent survey, the type of toilet facility was associated with diarrhea with an aOR of 0.79 (95% CI: 0.67-0.93) for improved facilities compared to unimproved ones. No associations with diarrhea were found for cooking place and water treatment across surveys.

## Neighborhood and household disparities

The random effect estimates from the multilevel regression analyses, specifically the ICC, as well as the MOR, indicated a trend in the clustering of diarrhea among children at the PSU and household levels (Tables 2 and 3). Both measures decreased between 1991 and 2006, indicating reduced disparities between clusters, but increased again in 2013 and further in 2022. Between-cluster differences were much more pronounced for households compared to PSUs.

**Table 2. Factors associated with recent diarrhea in children under 5 years of age in Yemen 1991 DHS and 2006 MICS.**

| Year | 1991 DHS | | | | | 2006 MICS | | | | |
|---|---|---|---|---|---|---|---|---|---|---|
| Characteristics | Univariable regression analysis | | | Multilevel regression analysis* | | Univariable regression analysis | | | Multilevel regression analysis* | |
| | OR† (95% CI†) | p-value | F test p-value‡ | aOR† (95% CI†) | p-value§ | OR† (95% CI†) | p-value | F test p-value‡ | aOR† (95% CI†) | p-value§ |
| **I. Individual-level factors** | | | | | | | | | | |
| Sex of child | | | | | | | | | | |
| Male | 1.00 | | 0.146 | 1.00 | | 1.00 | | 0.146 | 1.00 | |
| Female | 0.91 (0.81-1.03) | 0.146 | | 0.84 (0.67-1.05) | 0.123 | 0.90 (0.79-1.04) | 0.146 | | 0.83 (0.70-0.99) | 0.034 |
| Current age of child (Months) | | | | | | | | | | |
| 0-11 | 1.00 | | <0.001 | 1.00 | | 1.00 | | <0.001 | 1.00 | |
| 12-23 | 1.36 (1.17-1.59) | <0.001 | | 1.98 (1.44-2.72) | <0.001 | 1.35 (1.09-1.68) | 0.007 | | 1.62 (1.23-2.14) | 0.001 |
| 24-35 | 1.03 (0.87-1.21) | 0.744 | | 1.00 (0.73-1.37) | 0.996 | 1.02 (0.81-1.29) | 0.860 | | 1.02 (0.76-1.37) | 0.884 |
| 36-47 | 0.67 (0.57-0.80) | <0.001 | | 0.42 (0.30-0.59) | <0.001 | 0.69 (0.55-0.86) | 0.001 | | 0.66 (0.50-0.88) | 0.004 |
| 48-59 | 0.59 (0.49-0.71) | <0.001 | | 0.31 (0.22-0.44) | <0.001 | 0.53 (0.43-0.66) | <0.001 | | 0.46 (0.34-0.61) | <0.001 |
| Mother's education | | | | | | | | | | |
| None | 1.00 | | <0.001 | 1.00 | | 1.00 | | 0.991 | -- | -- |
| Basic | 0.51 (0.39-0.67) | <0.001 | | 0.69 (0.43-1.12) | 0.130 | 0.99 (0.83-1.18) | 0.892 | | -- | -- |
| Intermediate-Advanced | 0.37 (0.25-0.55) | <0.001 | | 0.83 (0.41-1.66) | 0.592 | 1.00 (0.76-1.30) | 0.977 | | -- | -- |
| **II. Household-level factors** | | | | | | | | | | |
| Place of residence | | | | | | | | | | |
| Urban | 1.00 | | <0.001 | 1.00 | | 1.00 | | 0.007 | 1.00 | |
| Rural | 1.61 (1.32-1.98) | <0.001 | | 0.99 (0.56-1.76) | 0.985 | 1.32 (1.08-1.61) | 0.007 | | 1.43 (1.02-2.00) | 0.040 |
| Region | | | | | | | | | | |
| South Yemen | 1.00 | | <0.001 | 1.00 | | 1.00 | | <0.001 | 1.00 | |
| North Yemen | 4.51 (3.56-5.70) | <0.001 | | 11.10 (6.74-18.27) | <0.001 | 1.65 (1.27-2.15) | <0.001 | | 1.98 (1.31-3.00) | 0.001 |
| No. of individuals in household | | | | | | | | | | |
| 1-5 | 1.00 | | 0.473 | -- | -- | 1.00 | | 0.148 | 1.00 | |
| 6-10 | 0.95 (0.81-1.12) | 0.544 | | -- | -- | 0.81 (0.67-0.99) | 0.035 | | 0.78 (0.61-1.00) | 0.047 |
| 11-15 | 0.88 (0.71-1.10) | 0.268 | | -- | -- | 0.77 (0.59-0.99) | 0.046 | | 0.71 (0.51-0.98) | 0.040 |
| >15 | 0.78 (0.55-1.11) | 0.172 | | -- | -- | 0.78 (0.59-1.04) | 0.095 | | 0.88 (0.61-1.27) | 0.496 |
| Wealth indexⁱ | | | | | | | | | | |
| Lowest | 1.00 | | <0.001 | 1.00 | | 1.00 | | 0.017 | 1.00 | |
| Second | 0.88 (0.66-1.16) | 0.355 | | 0.73 (0.45-1.21) | 0.226 | 1.06 (0.81-1.38) | 0.691 | | 1.06 (0.76-1.49) | 0.711 |

*(Continued)*

**Table 2.** (Continued)

| Year | 1991 DHS | | | | | 2006 MICS | | | | |
|---|---|---|---|---|---|---|---|---|---|---|
| Characteristics | Univariable regression analysis | | | Multilevel regression analysis* | | Univariable regression analysis | | | Multilevel regression analysis* | |
| | OR† (95% CI†) | p-value | F test p-value‡ | aOR† (95% CI†) | p-value§ | OR† (95% CI†) | p-value | F test p-value‡ | aOR† (95% CI†) | p-value§ |
| Middle | 0.79 (0.60-1.04) | 0.092 | | 0.57 (0.35-0.94) | 0.026 | 1.01 (0.78-1.30) | 0.962 | | 1.13 (0.81-1.60) | 0.463 |
| Fourth | 0.72 (0.54-0.95) | 0.020 | | 0.60 (0.36-0.99) | 0.045 | 0.90 (0.68-1.19) | 0.450 | | 1.18 (0.78-1.80) | 0.415 |
| Highest | 0.44 (0.33-0.58) | <0.001 | | 0.41 (0.22-0.75) | 0.004 | 0.67 (0.50-0.89) | 0.007 | | 0.87 (0.56-1.37) | 0.554 |
| Cooking place | | | | | | | | | | |
| House | 1.00 | | 0.006 | 1.00 | | 1.00 | | 0.873 | -- | -- |
| Separate building | 1.31 (1.10-1.56) | 0.003 | | 1.13 (0.82-1.57) | 0.448 | 1.04 (0.89-1.22) | 0.613 | | -- | -- |
| Other | 0.91 (0.62-1.33) | 0.614 | | 0.47 (0.23-1.00) | 0.050 | 1.14 (0.22-5.89) | 0.879 | | -- | -- |
| **_WASH-related factors_** | | | | | | | | | | |
| Source of drinking water¶ | | | | | | | | | | |
| Unimproved | 1.00 | | 0.126 | 1.00 | | 1.00 | | 0.181 | 1.00 | |
| Improved | 0.86 (0.71-1.04) | 0.126 | | 1.33 (0.86-2.06) | 0.199 | 0.88 (0.73-1.06) | 0.181 | | 0.98 (0.76-1.28) | 0.905 |
| Time to water source (Minutes) | | | | | | | | | | |
| On premises | 1.00 | | 0.002 | 1.00 | | 1.00 | | 0.660 | -- | -- |
| ≤30 | 1.21 (0.95-1.53) | 0.117 | | 1.01 (0.59-1.75) | 0.959 | 1.12 (0.88-1.44) | 0.362 | | -- | -- |
| >30 | 1.56 (1.21-2.02) | 0.001 | | 1.34 (0.71-2.53) | 0.366 | 1.04 (0.84-1.29) | 0.715 | | -- | -- |
| Type of toilet facility£ | | | | | | | | | | |
| Unimproved | 1.00 | | <0.001 | 1.00 | | 1.00 | | 0.352 | -- | -- |
| Improved | 0.48 (0.35-0.65) | <0.001 | | 0.79 (0.44-1.43) | 0.438 | 0.92 (0.77-1.10) | 0.352 | | -- | -- |
| Water treatment | | | | | | | | | | |
| No | 1.00 | | 0.861 | -- | -- | 1.00 | | 0.299 | -- | -- |
| Yes | 0.97 (0.71-1.34) | 0.861 | | -- | -- | 0.84 (0.60-1.17) | 0.299 | | -- | -- |
| Random effects estimates | | | | | | | | | | |
| PSU variance (95 CI %†) | NA | | | 0.87 (0.55-1.38) | | NA | | | 0.25 (0.14-0.44) | |
| Household variance (95 CI %†) | NA | | | 6.95 (5.49-9.03) | | NA | | | 1.24 (0.82-1.87) | |
| ICC** PSU (95 CI %†) | NA | | | 0.08 (0.05-0.11) | | NA | | | 0.05 (0.03-0.09) | |
| ICC** Household (95 CI %†) | NA | | | 0.71 (0.65-0.76) | | NA | | | 0.31 (0.24-0.39) | |
| MOR PSU (95 CI %†) | NA | | | 2.44 (1.94-2.94) | | NA | | | 1.60 (1.38-1.83) | |
| MOR Household (95 CI %†) | NA | | | 12.57 (8.62-16.52) | | NA | | | 2.89 (2.25-3.53) | |

Abbreviations: aOR, adjusted odds ratio; CI, confidence interval; DHS, demographic and health survey; ICC, intraclass correlation coefficient; MICS, multiple indicator cluster survey; MOR, median odds ratio; NA, not applicable; No., number; OR, odds ratio; PSU, primary sampling unit; WASH, Water/Sanitation/Hygiene.

*(Continued)*

**Table 2.** (Continued)

\* A three-level logistic regression model was used for each survey round, incorporating individual-level and household-level factors at the first level, household units at the second level, and PSUs at the third level. Detailed analyses illustrating the construction of the three-level logistic regression model for each round are provided in S2–S5 Tables.

† Estimates were calculated applying DHS or MICS sampling weights to account for the complex survey design of the DHS or MICS study.

‡ Covariates with p-value ≤0.2 in the univariable analysis were included in the multilevel analysis.

§ Covariates with p-value <0.05 in the multivariable analysis were considered as showing statistically significant evidence for an association with recent diarrhea.

¦ The wealth index for the 1991 survey was constructed following DHS guidelines, employing factor analysis. The selection of variables for this analysis was informed by those employed by the DHS in constructing the wealth index for the 2013 Yemen DHS. These variables encompassed household assets and amenities such as air conditioning, bicycle, blender, car, color television, dwelling type, electric fan, electricity, flooring material, gas or electric stove, motorcycle, number of sleeping rooms, primary source of drinking water, radio, refrigerator, sewing machine, telephone, television, type of toilet facility, vacuum cleaner, video player, washing machine, and water heater.

¶ In classifying water sources, "unimproved" refers to sources such as regular wells, unprotected water surfaces, rivers, tanker trucks, and containerized water, while "improved" refers to government and local network supplies, tube and pumped wells, rainwater, and bottled water.

£ In classifying toilet facilities, "unimproved" refers to pit latrines, flush toilets without sewer connections, open-drain toilets, street toilets, bucket toilets, shared facilities, and open defecation, while "improved" refers to flush toilets with piped sewer connections or septic tank systems.

\*\* Higher ICC values indicate a stronger clustering effect.

For PSUs, ICC was 0.08 in 1991, 0.05 in 2006, 0.06 in 2013, and 0.09 in 2022. Corresponding MOR estimates were 2.44, 1.60, 1.81, and 2.16, respectively.

For households, ICC was 0.71 in 1991, 0.31 in 2006, 0.53 in 2013, and 0.57 in 2022. Corresponding MOR estimates were 12.57, 2.89, 5.62, and 6.17, respectively.

To aid interpretation of these measures, the ICC of 0.09 for PSUs in 2022 indicates that 9% of the variation in diarrhea prevalence was attributable to differences between PSUs, reflecting modest regional variation.

In contrast, the household-level MOR of 6.17 in 2022 indicates that a child moving from a low-risk to a high-risk household would face more than sixfold higher odds of experiencing diarrhea, highlighting pronounced disparities at the household level.

### Sensitivity analyses

The sensitivity analyses incorporating birth weight, breastfeeding status, and health card possession for children under three confirmed the main study findings and revealed a similar clustering effect at the household level along with some clustering at the PSU level (S6 and S7 Tables).

Higher birth weight was generally inversely associated with diarrhea, although this association did not always reach statistical significance (S6 and S7 Tables). No consistent patterns were observed for the association between diarrhea and either breastfeeding or health card possession.

### Global diarrhea prevalence estimates

Global estimates for the pooled mean diarrhea prevalence showed a steady decline over time, decreasing from 21.6% (95% CI: 17.8-25.6%) in 1985–1989 to 10.1% (95% CI: 8.2-12.2%) in 2020–2024 (Table 4). A similar downward trend was observed in most WHO regions.

Between 1985–1994 and 2015–2024, the pooled mean prevalence decreased from 20.5% (95% CI: 17.6-23.6%) to 14.0% (95% CI: 12.8-15.4%) in AFR, 24.7% (95% CI: 23.3-26.1%) to 7.3% (95% CI: 2.9-13.4%) in EUR, 10.9% (95% CI: 7.2-15.2%) to 8.5% (95% CI: 6.5-10.7%) in SEAR, and 10.1% (95% CI: 9.5-10.8%) to 7.7% (95% CI: 4.6-11.7%) in WPR.

Meanwhile, it remained stable at 19.6% (95% CI: 15.1-24.5%) and 21.2% (95% CI: 20.2-22.3%) in AMR and at 17.7% (95% CI: 12.0-24.1%) and 18.5% (95% CI: 8.9-30.6%) in EMR (excluding Yemen). In contrast, diarrhea prevalence in Yemen showed an overall increasing trend over time (Fig 2).

**Table 3. Factors associated with recent diarrhea in children under 5 years of age in Yemen 2013 DHS and 2022 MICS.**

| Year | 2013 DHS | | | | | 2022 MICS | | | | |
|---|---|---|---|---|---|---|---|---|---|---|
| Characteristics | Univariable regression analysis | | | Multilevel regression analysis* | | Univariable regression analysis | | | Multilevel regression analysis* | |
| | OR† (95% CI†) | p-value | F test p-value‡ | aOR† (95% CI†) | p-value§ | OR† (95% CI†) | p-value | F test p-value‡ | aOR† (95% CI†) | p-value§ |
| **I. Individual-level factors** | | | | | | | | | | |
| Sex of child | | | | | | | | | | |
| Male | 1.00 | | 0.009 | 1.00 | | 1.00 | | 0.004 | 1.00 | |
| Female | 0.90 (0.83-0.97) | 0.009 | | 0.85 (0.75-0.96) | 0.010 | 0.90 (0.83-0.97) | 0.004 | | 0.84 (0.75-0.95) | 0.006 |
| Current age of child (Months) | | | | | | | | | | |
| 0-11 | 1.00 | | <0.001 | 1.00 | | 1.00 | | <0.001 | 1.00 | |
| 12-23 | 1.53 (1.36-1.73) | <0.001 | | 1.99 (1.64-2.42) | <0.001 | 1.38 (1.23-1.55) | <0.001 | | 1.73 (1.42-2.11) | <0.001 |
| 24-35 | 0.83 (0.73-0.95) | 0.007 | | 0.80 (0.65-0.98) | 0.032 | 0.84 (0.76-0.94) | 0.003 | | 0.82 (0.68-0.98) | 0.033 |
| 36-47 | 0.59 (0.52-0.66) | <0.001 | | 0.41 (0.33-0.50) | <0.001 | 0.69 (0.60-0.78) | <0.001 | | 0.55 (0.45-0.68) | <0.001 |
| 48-59 | 0.37 (0.32-0.43) | <0.001 | | 0.19 (0.15-0.24) | <0.001 | 0.57 (0.51-0.64) | <0.001 | | 0.38 (0.31-0.46) | <0.001 |
| Mother's education | | | | | | | | | | |
| None | 1.00 | | 0.170 | 1.00 | | 1.00 | | 0.027 | 1.00 | |
| Basic | 0.87 (0.73-1.03) | 0.100 | | 1.01 (0.79-1.31) | 0.910 | 1.01 (0.90-1.13) | 0.921 | | 1.19 (1.00-1.41) | 0.053 |
| Intermediate-Advanced | 1.02 (0.91-1.15) | 0.677 | | 1.19 (1.00-1.42) | 0.052 | 0.87 (0.78-0.98) | 0.022 | | 1.21 (1.01-1.45) | 0.042 |
| **II. Household-level factors** | | | | | | | | | | |
| Place of residence | | | | | | | | | | |
| Urban | 1.00 | | 0.014 | 1.00 | | 1.00 | | <0.001 | 1.00 | |
| Rural | 1.23 (1.04-1.44) | 0.014 | | 1.09 (0.85-1.41) | 0.491 | 1.34 (1.19-1.51) | <0.001 | | 0.69 (0.54-0.89) | 0.005 |
| Region | | | | | | | | | | |
| South Yemen | 1.00 | | <0.001 | 1.00 | | 1.00 | | <0.001 | 1.00 | |
| North Yemen | 1.89 (1.66-2.16) | <0.001 | | 2.31 (1.86-2.88) | <0.001 | 3.41 (2.98-3.91) | <0.001 | | 4.60 (3.61-5.86) | <0.001 |
| No. of individuals in household | | | | | | | | | | |
| 1-5 | 1.00 | | 0.112 | 1.00 | | 1.00 | | <0.001 | 1.00 | |
| 6-10 | 0.88 (0.79-0.97) | 0.015 | | 0.81 (0.70-0.95) | 0.009 | 1.03 (0.93-1.14) | 0.541 | | 1.04 (0.89-1.20) | 0.646 |
| 11-15 | 0.91 (0.79-1.04) | 0.167 | | 0.77 (0.62-0.95) | 0.014 | 0.87 (0.75-1.01) | 0.063 | | 0.77 (0.64-0.94) | 0.009 |
| >15 | 0.90 (0.72-1.12) | 0.345 | | 0.91 (0.69-1.20) | 0.505 | 0.66 (0.55-0.81) | <0.001 | | 0.68 (0.52-0.88) | 0.004 |

*(Continued)*

**Table 3.** (Continued)

| Year | 2013 DHS | | | | | 2022 MICS | | | | |
|---|---|---|---|---|---|---|---|---|---|---|
| Characteristics | Univariable regression analysis | | | Multilevel regression analysis* | | Univariable regression analysis | | | Multilevel regression analysis* | |
| | OR† (95% CI†) | p-value | F test p-value‡ | aOR† (95% CI†) | p-value§ | OR† (95% CI†) | p-value | F test p-value‡ | aOR† (95% CI†) | p-value§ |
| Wealth index | | | | | | | | | | |
| Lowest | 1.00 | | <0.001 | 1.00 | | 1.00 | | <0.001 | 1.00 | |
| Second | 1.06 (0.92-1.22) | 0.407 | | 1.07 (0.87-1.32) | 0.510 | 0.93 (0.80-1.08) | 0.356 | | 0.91 (0.73-1.12) | 0.367 |
| Middle | 1.03 (0.88-1.20) | 0.713 | | 1.19 (0.93-1.52) | 0.171 | 0.74 (0.63-0.86) | <0.001 | | 0.67 (0.51-0.87) | 0.003 |
| Fourth | 0.91 (0.76-1.08) | 0.276 | | 1.03 (0.76-1.39) | 0.854 | 0.66 (0.56-0.78) | <0.001 | | 0.57 (0.41-0.79) | 0.001 |
| Highest | 0.70 (0.59-0.84) | <0.001 | | 0.72 (0.51-1.01) | 0.059 | 0.38 (0.32-0.45) | <0.001 | | 0.31 (0.21-0.46) | <0.001 |
| Cooking place | | | | | | | | | | |
| House | 1.00 | | 0.959 | -- | -- | 1.00 | | <0.001 | 1.00 | |
| Separate building | 1.01 (0.91-1.14) | 0.802 | | -- | -- | 1.40 (1.26-1.55) | <0.001 | | 1.07 (0.91-1.26) | 0.403 |
| Other | 1.02 (0.82-1.27) | 0.858 | | -- | -- | 1.16 (0.58-2.32) | 0.684 | | 0.93 (0.33-2.62) | 0.886 |
| *WASH-related factors* | | | | | | | | | | |
| Source of drinking water⌐ | | | | | | | | | | |
| Unimproved | 1.00 | | <0.001 | 1.00 | | 1.00 | | <0.001 | 1.00 | |
| Improved | 0.77 (0.69-0.86) | <0.001 | | 0.84 (0.71-0.99) | 0.040 | 0.74 (0.64-0.84) | <0.001 | | 1.11 (0.91-1.36) | 0.282 |
| Time to water source (Minutes) | | | | | | | | | | |
| On premises | 1.00 | | <0.001 | 1.00 | | 1.00 | | <0.001 | 1.00 | |
| ≤30 | 1.53 (1.21-1.93) | <0.001 | | 1.51 (1.08-2.11) | 0.015 | 1.61 (1.40-1.85) | <0.001 | | 1.38 (1.12-1.70) | 0.002 |
| >30 | 1.47 (1.24-1.73) | <0.001 | | 1.66 (1.31-2.10) | <0.001 | 1.59 (1.39-1.82) | <0.001 | | 1.30 (1.04-1.64) | 0.023 |
| Type of toilet facility¶ | | | | | | | | | | |
| Unimproved | 1.00 | | 0.028 | 1.00 | | 1.00 | | <0.001 | 1.00 | |
| Improved | 0.88 (0.79-0.99) | 0.028 | | 0.96 (0.80-1.16) | 0.675 | 0.67 (0.60-0.75) | <0.001 | | 0.79 (0.67-0.93) | 0.005 |
| Water treatment | | | | | | | | | | |
| No | 1.00 | | 0.217 | -- | -- | 1.00 | | 0.296 | -- | -- |
| Yes | 1.10 (0.95-1.27) | 0.217 | | -- | -- | 1.08 (0.93-1.26) | 0.296 | | -- | -- |
| Random effects estimates | | | | | | | | | | |
| PSU variance (95 CI %†) | NA | | | 0.39 (0.28-0.54) | | NA | | | 0.65 (0.51-0.84) | |
| Household variance (95 CI %†) | NA | | | 3.28 (2.61-4.12) | | NA | | | 3.64 (3.00-4.41) | |
| ICC£ PSU (95 CI %†) | NA | | | 0.06 (0.04-0.07) | | NA | | | 0.09 (0.07-0.11) | |
| ICC£ Household (95 CI %†) | NA | | | 0.53 (0.47-0.58) | | NA | | | 0.57 (0.52-0.61) | |

*(Continued)*

## PLOS Neglected Tropical Diseases

**Table 3.** (Continued)

| Year | 2013 DHS | | | | | 2022 MICS | | | | |
|---|---|---|---|---|---|---|---|---|---|---|
| **Characteristics** | **Univariable regression analysis** | | | **Multilevel regression analysis\*** | | **Univariable regression analysis** | | | **Multilevel regression analysis\*** | |
| | **OR[†] (95% CI[†])** | **p-value** | **F test p-value[‡]** | **aOR[†] (95% CI[†])** | **p-value[§]** | **OR[†] (95% CI[†])** | **p-value** | **F test p-value[‡]** | **aOR[†] (95% CI[†])** | **p-value[§]** |
| MOR PSU (95 CI %[†]) | NA | | | 1.81 (1.63-1.99) | | NA | | | 2.16 (1.95-2.37) | |
| MOR Household (95 CI %[†]) | NA | | | 5.62 (4.51-6.73) | | NA | | | 6.17 (5.09-7.25) | |

Abbreviations: aOR, adjusted odds ratio; CI, confidence interval; DHS, demographic and health survey; ICC, intraclass correlation coefficient; MICS, multiple indicator cluster survey; MOR, median odds ratio; NA, not applicable; No., number; OR, odds ratio; PSU, primary sampling unit; WASH, Water/Sanitation/Hygiene.

\* A three-level logistic regression model was used for each survey round, incorporating individua-level and household-level factors at the first level, household units at the second level, and PSUs at the third level. Detailed analyses illustrating the construction of the three-level logistic regression model for each round are provided in S2–S5 Tables.

† Estimates were calculated applying DHS or MICS sampling weights to account for the complex survey design of the DHS or MICS study.

‡ Covariates with p-value ≤0.2 in the univariable analysis were included in the multilevel analysis.

§ Covariates with p-value <0.05 in the multivariable analysis were considered as showing statistically significant evidence for an association with recent diarrhea.

¦ In classifying water sources, "unimproved" refers to sources such as regular wells, unprotected water surfaces, rivers, tanker trucks, and containerized water, while "improved" refers to government and local network supplies, tube and pumped wells, rainwater, and bottled water.

¶ In classifying toilet facilities, "unimproved" refers to pit latrines, flush toilets without sewer connections, open-drain toilets, street toilets, bucket toilets, shared facilities, and open defecation, while "improved" refers to flush toilets with piped sewer connections or septic tank systems.

£ Higher ICC values indicate a stronger clustering effect.

In all meta-analyses, $I^2$ was greater than 98%, indicating that the variation in prevalence across surveys was primarily due to true differences rather than random chance.

## Discussion

This study assessed prevalence, trend, and risk factors associated with childhood diarrhea in Yemen, utilizing data from four nationally representative population-based surveys spanning three decades. The findings indicated a persistent, if not increasing, burden of diarrhea among Yemeni children, with prevalence exceeding 30% over the last decades and approaching 40% in 2022, in contrast to the global declining trend. Yemen's diarrhea prevalence consistently remained more than double the global estimate and surpassed all regional estimates across the different time periods.

Identified risk factors included the child's age, residence in North Yemen, lack of access to water on premises, unimproved toilet facilities, smaller household size, and low wealth quintile. Disparities in diarrhea were much more pronounced between households than across neighborhoods, highlighting the pivotal role of household-level socioeconomic factors and living conditions in determining diarrhea risk.

These findings underscore that Yemen's persistent burden of diarrhea is rooted in the country's longstanding challenges. Chronic water shortages and inadequate access to improved sanitation have long contributed to the endemicity of intestinal parasitic infections [56–58]. These issues have been further exacerbated by ongoing political unrest.

While global and regional declines in diarrhea have been mainly driven by improvements in WASH [59,60], particularly in regions like Africa [61] (Table 4), countries facing political instability or natural disasters, such as those in AMR and EMR, have often experienced stagnant or even increasing prevalence [62,63]. Yemen exemplifies this trend. Despite initial improvements, the country's protracted conflict, economic instability, and widespread poverty have severely compromised

**Table 4. Meta-analyses estimating the pooled mean diarrhea prevalence in children under 5 years of age in countries with available DHS, for each WHO region and globally.**

| Diarrhea prevalence | Countries | Sample | | Pooled mean diarrhea prevalence | | Heterogeneity measure |
|---|---|---|---|---|---|---|
| | N | Total | Recent diarrhea | (%) | 95% CI | I²* (%; 95% CI) |
| **AFR** | | | | | | |
| 1985-1994 | 21 | 96,232 | 19,349 | 20.5 | 17.6-23.6 | 99.2 (99.1-99.3) |
| 1995-2004 | 33 | 198,292 | 35,071 | 17.1 | 15.7-18.5 | 98.5 (98.3-98.7) |
| 2005-2014 | 57 | 505,951 | 76,341 | 15.3 | 14.2-16.4 | 99.3 (99.2-99.4) |
| 2015-2024 | 33 | 328,276 | 47,144 | 14.0 | 12.8-15.4 | 99.2 (99.1-99.4) |
| **AMR** | | | | | | |
| 1985-1994 | 14 | 55,071 | 11,340 | 19.6 | 15.1-24.5 | 99.4 (99.3-99.5) |
| 1995-2004 | 16 | 110,983 | 18,848 | 16.5 | 14.8-18.3 | 98.4 (98.0-98.7) |
| 2005-2014 | 16 | 139,228 | 22,416 | 16.1 | 14.1-18.2 | 98.9 (98.7-99.1) |
| 2015-2024 | 1 | 5,867 | 1,244 | 21.2† | 20.2-22.3 | -- |
| **EMR‡** | | | | | | |
| 1985-1994 | 7 | 42,678 | 7,504 | 17.7 | 12.0-24.1 | 99.7 (99.6-99.7) |
| 1995-2004 | 6 | 45,234 | 6,255 | 14.2 | 10.7-18.1 | 99.4 (99.2-99.5) |
| 2005-2014 | 7 | 77,453 | 12,791 | 16.4 | 12.9-20.3 | 99.5 (99.3-99.6) |
| 2015-2024 | 3 | 49,558 | 11,477 | 18.5 | 8.9-30.6 | 99.9 (99.9-99.9) |
| **EUR** | | | | | | |
| 1985-1994 | 1 | 3,532 | 872 | 24.7† | 23.3-26.1 | -- |
| 1995-2004 | 4 | 9,541 | 1,390 | 12.0 | 3.5-24.6 | 99.7 (99.6-99.8) |
| 2005-2014 | 7 | 17,205 | 1,765 | 9.5 | 6.4-13.0 | 98.4 (97.8-98.9) |
| 2015-2024 | 3 | 10,509 | 1,050 | 7.3 | 2.9-13.4 | 99.1 (98.5-99.4) |
| **SEAR** | | | | | | |
| 1985-1994 | 4 | 36,672 | 4,177 | 10.9 | 7.2-15.2 | 98.6 (97.7-99.1) |
| 1995-2004 | 6 | 54,952 | 5,827 | 10.1 | 6.7-14.2 | 99.4 (99.2-99.5) |
| 2005-2014 | 10 | 130,449 | 13,621 | 9.9 | 7.3-12.8 | 99.5 (99.3-99.6) |
| 2015-2024 | 9 | 509,961 | 43,127 | 8.5 | 6.5-10.7 | 99.5 (99.3-99.6) |
| **WPR** | | | | | | |
| 1985-1994 | 1 | 8,512 | 860 | 10.1† | 9.5-10.8 | -- |
| 1995-2004 | 3 | 21,325 | 2,635 | 11.9 | 6.2-19.2 | 99.6 (99.4-99.7) |
| 2005-2014 | 5 | 35,107 | 4,585 | 12.6 | 8.8-16.9 | 99.3 (99.0-99.5) |
| 2015-2024 | 4 | 34,580 | 2,834 | 7.7 | 4.6-11.7 | 99.4 (99.2-99.6) |
| **Global‡** | | | | | | |
| 1985-1989 | 23 | 89,768 | 20,274 | 21.6 | 17.8-25.6 | 99.5 (99.4-99.5) |
| 1990-1994 | 25 | 152,929 | 23,828 | 16.4 | 14.2-18.6 | 99.2 (99.1-99.3) |
| 1995-1999 | 28 | 163,516 | 25,565 | 15.7 | 13.8-17.7 | 99.2 (99.1-99.3) |
| 2000-2004 | 40 | 276,811 | 44,461 | 15.3 | 13.7-17.0 | 99.3 (99.2-99.3) |
| 2005-2009 | 47 | 394,030 | 55,138 | 14.1 | 12.7-15.6 | 99.4 (99.3-99.4) |
| 2010-2014 | 55 | 511,363 | 76,381 | 14.6 | 13.3-15.9 | 99.4 (99.3-99.4) |
| 2015-2019 | 43 | 840,438 | 96,259 | 13.0 | 11.4-14.7 | 99.8 (99.8-99.8) |
| 2020-2024 | 10 | 98,313 | 10,617 | 10.1 | 8.2-12.2 | 99.1 (99.9-99.3) |

Abbreviations: AFR, African Region; AMR, Region of the Americas; CI, confidence interval; EMR, Eastern Mediterranean Region; EUR, European Region; SEAR, South-East Asia Region; WHO, World Health Organization; WPR, Western Pacific Region.

\* I²: a measure assessing the magnitude of between-study variation that is due to differences in effect size (here, diarrhea prevalence) across studies rather than chance.

† Point estimate as only one study was available.

‡ Excluding Yemen.

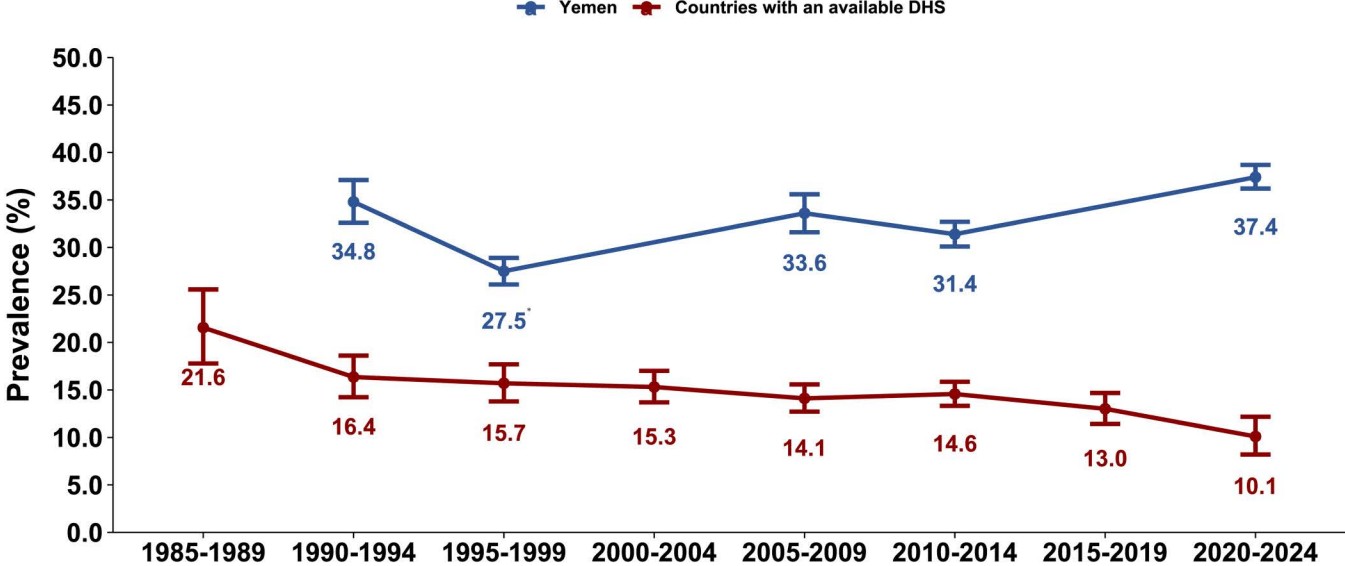

**Fig 2. Prevalence of recent diarrhea in children under 5 years of age in Yemen compared to pooled 5-year interval data from countries with available DHS.** Abbreviation: DHS, demographic and health survey. *Although the complete dataset for Yemen's 1997 DHS was restricted, the diarrhea prevalence estimate for this survey was available and included in the comparative analysis.

public health infrastructure and services [11,63–65], as reflected in the increasing disparities between neighborhoods and households over the last decade.

This deterioration in health infrastructure is starkly illustrated by Yemen's immunization program. Despite rotavirus vaccination being introduced in 2012 with initial success in reducing hospitalizations [66], coverage has remained consistently low at around 60% [67]—far below the WHO target of 90% [68]. The suspension of outreach immunization activities in North Yemen during the conflict further compromised vaccine access, particularly in remote areas [69]. With nearly half of health facilities non-functional and over 30% no longer providing vaccination services [69], these setbacks have left children increasingly vulnerable to preventable diseases such as diarrhea.

An age-related pattern was evident across all surveys. The odds of diarrhea were highest in children aged 12–23 months and decreased progressively with increasing age, consistent with findings of studies from other countries [24,26,27,29,31,32,70,71]. The increased vulnerability in the 12–23-month age group likely stems from greater exposure to pathogens during weaning, as children transition from breast milk to solid foods and come into contact with unsterilized feeding equipment or contaminated food and water [26,27,70]. The risk of diarrhea subsequently declines as natural immunity builds with age [26,70].

Females had, on average, a 15% lower odds of diarrhea compared to males in recent surveys. Similar findings have been reported in studies from other countries [28,70,72]. The reasons for this difference remain uncertain but may involve cultural practices, where male children, spending more time outdoors or accompanying their fathers, are exposed to environmental pathogens more frequently [28,70]. Inherent biological sex differences may also influence susceptibility to diarrhea [28,70,73]. Further research is warranted to clarify this effect.

Children in North Yemen consistently exhibited higher odds of diarrhea compared to those in the South. In 1991, the odds were more than 11 times higher in the North, underscoring deep-rooted structural and public health disparities stemming from the period of separate governance prior to unification [74,75], and further sustained by ongoing political fragmentation [76]. North Yemen also experiences more frequent rainfall and flooding, which increases exposure to

waterborne pathogens [56,57]. The region's mountainous terrain further complicates access to healthcare services and clean water, and exacerbates the risk of landslides that can damage household infrastructure [57,77,78].

The current conflict has added another layer of complexity to these geographic health patterns by causing large-scale population displacement. In the most recent survey, a counterintuitive finding emerged: diarrhea odds were 31% lower in rural areas than in urban ones. This may reflect rural-to-urban migration, with displaced populations settling in over-crowded urban slums with poor water, sanitation, and healthcare services [79,80]. Simultaneously, reverse migration has strained rural communities hosting internally displaced persons, overwhelming their limited infrastructure [81].

Households without on-premises water access were associated with higher odds of diarrhea, likely due to limited ability to maintain essential hygiene practices such as cleaning, toilet flushing, and handwashing—simple yet effective measures for preventing diarrhea [82,83]. The need to store drinking water for extended periods, due to the distance from water sources, and repeated extraction with unclean utensils or hands, further increases the risk of microbial contamination [84]. The use of water sources such as standpipes or public taps may also contribute to the risk of contamination [27].

Notably, the associations with distance to water sources and use of unimproved toilet facilities only emerged in recent surveys, perhaps reflecting the impact of the ongoing conflict on infrastructure and the resulting internal displacement. This conflict has forced many individuals into suboptimal living conditions, exacerbating the risk of diarrhea [11,63,85].

Children from wealthier households were less likely to experience diarrhea, aligning with previous research indicating wealthier households as typically having better access to improved water, sanitation, hygiene facilities, hygiene education, and health services [26,86–89]. Larger households generally had lower odds of diarrhea, possibly benefiting from more caregivers or working adults contributing to childcare and household resources [90].

## Strengths and limitations

This study was based on nationally representative data from four DHS and MICS surveys conducted over three decades in Yemen. The standardized methodologies employed across these surveys ensured comparability, allowing for an analysis of long-term trends. By using a three-level logistic regression model, the study was able to assess associations and variability at both neighborhood and household levels, providing a nuanced understanding of risk factors. The study also included a comparison to global levels and trends through supplementary meta-analyses of global and regional DHS data.

However, the study has some limitations. Data were collected through self-reports from mothers or caregivers, potentially introducing recall or social desirability bias. Nonetheless, this is not likely given the short two-week recall period and the non-stigmatizing nature of diarrheal cases. The cross-sectional design of the surveys limited causal inference, allowing only the identification of associations. The use of two different types of surveys restricted the inclusion of certain variables for specific age groups in the main analysis, such as birthweight or breastfeeding status. However, sensitivity analyses incorporating these factors suggested that this is not likely to have affected the results.

Some variables, such as 'ever breastfed', were too broadly defined to provide meaningful associations. Other child health indicators, including vaccination status and dietary diversity, were excluded from the main models due to either absence in certain survey rounds or data quality concerns.

Although real-time surveillance data from WHO's electronic Disease Early Warning System (eDEWS) could have provided additional insights, access was not possible due to account restrictions. Furthermore, previous studies have highlighted several limitations of eDEWS data for longitudinal analysis in Yemen, including delays in case verification, increasing dissemination lags, reliance on manual data entry, and incomplete data coverage [91].

## Conclusions for action

The findings indicates a persistent public health challenge in Yemen, with childhood diarrhea prevalence remaining high—and worsening in recent years—in stark contrast to global trends of improvement. The study underscores the dominant influence of household-level socioeconomic conditions and access to basic amenities, such as water and sanitation, in

shaping diarrhea risk. These factors appear to have a greater impact than broader neighborhood-level influences, highlighting the importance of household-targeted interventions.

Improving water and sanitation infrastructure is urgently needed across Yemen. In rural areas, mobile interventions such as water trucking and latrine construction may offer the most practical solutions, while urban areas could benefit from rehabilitating deteriorating infrastructure and establishing community water points where safe water storage and sanitation facilities are lacking. Complementary public health initiatives—including hygiene promotion, exclusive breastfeeding, safe weaning practices, and rotavirus vaccination—can further reduce diarrhea prevalence and support Yemen's progress toward the SDGs.

Targeted efforts should prioritize high-burden areas, particularly in North Yemen, and be implemented in close coordination with local non-governmental organizations, international partners, and existing public health systems. Ongoing monitoring and assessment of disparities will be essential for guiding interventions and promoting equitable health outcomes.

## Supporting information

**S1 Table. Strengthening the reporting of observational studies in epidemiology (STROBE) checklist.**
(DOCX)

**S2 Table. Multilevel logistic regression analyses investigating associations with recent diarrhea in children under 5 years of age in Yemen 1991 DHS.**
(DOCX)

**S3 Table. Multilevel logistic regression analyses investigating associations with recent diarrhea in children under 5 years of age in Yemen 2006 MICS.**
(DOCX)

**S4 Table. Multilevel logistic regression analyses investigating associations with recent diarrhea in children under 5 years of age in Yemen 2013 DHS.**
(DOCX)

**S5 Table. Multilevel logistic regression analyses investigating associations with recent diarrhea in children under 5 years of age in Yemen 2022 MICS.**
(DOCX)

**S6 Table. Sensitivity analyses. Regression models investigating additional factors potentially associated with recent diarrhea (birthweight, health card possession, and breastfeeding) in the subgroup of children under 3 years of age, for whom this information was available, in Yemen 1991 DHS and 2006 MICS.**
(DOCX)

**S7 Table. Sensitivity analyses. Regression models investigating additional factors potentially associated with recent diarrhea (birthweight, health card possession, and breastfeeding) in the subgroup of children under 3 years of age, for whom this information was available, in Yemen 2013 DHS and 2022 MICS.**
(DOCX)

## Author contributions

**Conceptualization:** Zahir M. Tag.

**Formal analysis:** Zahir M. Tag, Laith J. Abu-Raddad, Hiam Chemaitelly.

**Funding acquisition:** Laith J. Abu-Raddad.

**Investigation:** Zahir M. Tag.

**Methodology:** Zahir M. Tag, Laith J. Abu-Raddad, Hiam Chemaitelly.

**Supervision:** Hiam Chemaitelly.

**Validation:** Zahir M. Tag.

**Visualization:** Zahir M. Tag.

**Writing – original draft:** Zahir M. Tag, Hiam Chemaitelly.

**Writing – review & editing:** Zahir M. Tag, Laith J. Abu-Raddad, Hiam Chemaitelly.

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
