## [Decision Letter · Decision Letter 0]

PNTD-D-24-01198Diarrhea in Yemeni Children Under Five: A Multi-Level Analysis of Population-Based Surveys, 1991-2022PLOS Neglected Tropical Diseases Dear Dr. Tag, Thank you for submitting your manuscript to PLOS Neglected Tropical Diseases. After careful consideration, we feel that it has merit but does not fully meet PLOS Neglected Tropical Diseases's publication criteria as it currently stands. Therefore, we invite you to submit a revised version of the manuscript that addresses the points raised during the review process. Please submit your revised manuscript within 30 days Jul 27 2025 11:59PM. If you will need more time than this to complete your revisions, please reply to this message or contact the journal office at plosntds@plos.org. Please include the following items when submitting your revised manuscript:* A rebuttal letter that responds to each point raised by the editor and reviewer(s). You should upload this letter as a separate file labeled 'Response to Reviewers '. This file does not need to include responses to any formatting updates and technical items listed in the 'Journal Requirements' section below.* A marked-up copy of your manuscript that highlights changes made to the original version. You should upload this as a separate file labeled 'Revised Manuscript with Track Changes '.* An unmarked version of your revised paper without tracked changes. You should upload this as a separate file labeled 'Manuscript '. If you would like to make changes to your financial disclosure, competing interests statement, or data availability statement, please make these updates within the submission form at the time of resubmission. Guidelines for resubmitting your figure files are available below the reviewer comments at the end of this letter. We look forward to receiving your revised manuscript. Kind regards, Qu Cheng, Ph.D.Section EditorPLOS Neglected Tropical Diseases Qu ChengSection EditorPLOS Neglected Tropical Diseases

Shaden Kamhawi

co-Editor-in-Chief

Paul Brindley

co-Editor-in-Chief

 **Journal Requirements:**

1) Please upload all main figures as separate Figure files in .tif or .eps format. For more information about how to convert and format your figure files please see our guidelines: 

2) We have noticed that you have uploaded Supporting Information files, but you have not included a list of legends. Please add a full list of legends for your Supporting Information files after the references list.

3) Please amend your detailed Financial Disclosure statement. This is published with the article. It must therefore be completed in full sentences and contain the exact wording you wish to be published. State the initials, alongside each funding source, of each author to receive each grant. For example: "This work was supported by the National Institutes of Health (####### to AM; ###### to CJ) and the National Science Foundation (###### to AM).".

 **Reviewers' comments:** Reviewer's Responses to Questions

**Key Review Criteria Required for Acceptance?**

**Methods**

-Are the objectives of the study clearly articulated with a clear testable hypothesis stated?

-Is the study design appropriate to address the stated objectives?

-Is the population clearly described and appropriate for the hypothesis being tested?

-Is the sample size sufficient to ensure adequate power to address the hypothesis being tested?

-Were correct statistical analysis used to support conclusions?

-Are there concerns about ethical or regulatory requirements being met?

Reviewer #1: Manuscript Title;

Diarrhea in Yemeni Children Under Five: A Multi-Level Analysis of Population-Based Surveys, 1991-2022

Manuscript Number: PNTD-D-24-01198

Authors; Zahir M. Tag, , Laith J. Abu-Raddad, , and Hiam Chemaitelly,

Reviewer; Mohamed A Daw, MD, MPS, Pharm D, PhD, Mrcpath(1A), FTCDI, Faculty of Medicine, University of Tripoli –Libya

Parachute research;

Though I have gone through the whole MS, I did no see any Yamani institute or even participant was involved in this study. This raise a major question regarding the gaining and reliability of the data involved in the study, which raise the issue of Parachute research.

Recommendation

Reject ; or resubmit based on;

The authors should include a person or a Yamani institute who should be involved in the study

Reviewer #2: (No Response)

Reviewer #3: 1-Yes the study’s objectives are clearly stated. The authors aim to assess the prevalence, trends, and determinants of diarrhea among children under five in Yemen across a 30 year span. While the manuscript doesn’t formally state a hypothesis in the traditional “if-then”format, the analytical approach is driven by clear research questions focusing on associations between individual/household factors and diarrhea risk, making the investigation testable and hypothesis oriented in practice

2-Is the study design appropriate to address the stated objectives?

Absolutely. The use of nationally representative, cross sectional datasets (DHS and MICS) is well suited for identifying temporal trends and population level associations. The three level random effects and logistic regression is appropriate given the nested nature of the data (individuals within households within neighborhoods). Including both household and neighborhood clustering effects strengthens the credibility of the findings.

3-Yes, the study population is well defined. The authors used data from four large national surveys, covering over 45,000 children under five across multiple decades. The sampling methodology (two-stage cluster design) is robust and clearly explained, ensuring representativeness across different regions and socioeconomic backgrounds.

4-The sample sizes for each survey year are more than adequate to detect meaningful associations, with the 2022 MICS including nearly 20,000 children. This large sample ensures sufficient statistical power, especially given the multilevel structure of the analysis

5-The authors applied a thoughtful and thorough statistical approach. The three level multivariable logistic regression model is well justified and executed, accounting for clustering at the household and PSU levels. Model fit comparisons (via AIC/BIC), checks for multicollinearity (VIF), and the use of sensitivity analyses all reflect sound statistical practice. The supplementary global metaanalysis comparing Yemen to other regions further contextualizes the findings

6-There is no major concerns. The authors correctly note that ethical approval was not required due to the use of publicly available, de identified DHS and MICS datasets. They also appropriately acknowledge the sources of funding and confirm the absence of competing interests, which adds transparency

Reviewer #4: The study would benefit from incorporating real-time surveillance data from WHO’s Electronic Disease Early Warning System (eDEWS), which provides weekly updates on disease trends, including diarrhea outbreaks. Using current surveillance data would help validate the long-term trends identified and reflect the current situation more accurately.

**Results**

-Does the analysis presented match the analysis plan?

-Are the results clearly and completely presented?

-Are the figures (Tables, Images) of sufficient quality for clarity?

Reviewer #1: Manuscript Title;

Diarrhea in Yemeni Children Under Five: A Multi-Level Analysis of Population-Based Surveys, 1991-2022

Manuscript Number: PNTD-D-24-01198

Authors; Zahir M. Tag, , Laith J. Abu-Raddad, , and Hiam Chemaitelly,

Reviewer; Mohamed A Daw, MD, MPS, Pharm D, PhD, Mrcpath(1A), FTCDI, Faculty of Medicine, University of Tripoli –Libya

Parachute research;

Though I have gone through the whole MS, I did no see any Yamani institute or even participant was involved in this study. This raise a major question regarding the gaining and reliability of the data involved in the study, which raise the issue of Parachute research.

Recommendation

Reject ; or resubmit based on;

The authors should include a person or a Yamani institute who should be involved in the study

Reviewer #2: (No Response)

Reviewer #3: Yes, the analysis closely follows the plan outlined in the Methods section. The authors clearly indicate their intention to use multi-level logistic regression to examine individual and household level risk factors, accounting for clustering at the household and neighborhood (PSU) levels. This is consistently applied across all four survey rounds. The statistical techniques used including univariable screening, model selection via AIC/BIC, and the use of ICC and MOR to assess clustering are appropriate and align well with what was described up front. The inclusion of a sensitivity analysis and a global metaanalysis to provide context was also part of the methods and is well integrated into the final analysis.

Reviewer #4: The results section discusses random-effects estimates (ICC & MOR) but lacks sufficient interpretation for non-statistical readers. Consider explaining ICC and MOR in more practical terms, particularly in relation to household vs. neighborhood disparities.

**Conclusions**

-Are the conclusions supported by the data presented?

-Are the limitations of analysis clearly described?

-Do the authors discuss how these data can be helpful to advance our understanding of the topic under study?

-Is public health relevance addressed?

Reviewer #1: Manuscript Title;

Diarrhea in Yemeni Children Under Five: A Multi-Level Analysis of Population-Based Surveys, 1991-2022

Manuscript Number: PNTD-D-24-01198

Authors; Zahir M. Tag, , Laith J. Abu-Raddad, , and Hiam Chemaitelly,

Reviewer; Mohamed A Daw, MD, MPS, Pharm D, PhD, Mrcpath(1A), FTCDI, Faculty of Medicine, University of Tripoli –Libya

Parachute research;

Though I have gone through the whole MS, I did no see any Yamani institute or even participant was involved in this study. This raise a major question regarding the gaining and reliability of the data involved in the study, which raise the issue of Parachute research.

Recommendation

Reject ; or resubmit based on;

The authors should include a person or a Yamani institute who should be involved in the study

Reviewer #2: (No Response)

Reviewer #3: Yes, the conclusions are well supported by the data. The findings from the multilevel regression models, trend analyses, and sensitivity checks all align with the central claims made by the authors particularly regarding the persistence and worsening burden of childhood diarrhea in Yemen, the importance of household level factors, and the growing disparities over time. The statistical associations are clearly presented, and the interpretation of adjusted odds ratios is appropriately cautious and grounded in the data.

Reviewer #4: The study identifies risk factors but provides limited recommendations for policy action. Expand on potential WASH interventions: targeted sanitation programs, hygiene education, water infrastructure development, etc.

Discuss rural vs. urban intervention strategies to tailor solutions effectively.

**Editorial and Data Presentation Modifications?**

Reviewer #1: Manuscript Title;

Diarrhea in Yemeni Children Under Five: A Multi-Level Analysis of Population-Based Surveys, 1991-2022

Manuscript Number: PNTD-D-24-01198

Authors; Zahir M. Tag, , Laith J. Abu-Raddad, , and Hiam Chemaitelly,

Reviewer; Mohamed A Daw, MD, MPS, Pharm D, PhD, Mrcpath(1A), FTCDI, Faculty of Medicine, University of Tripoli –Libya

Parachute research;

Though I have gone through the whole MS, I did no see any Yamani institute or even participant was involved in this study. This raise a major question regarding the gaining and reliability of the data involved in the study, which raise the issue of Parachute research.

Recommendation

Reject ; or resubmit based on;

The authors should include a person or a Yamani institute who should be involved in the study

Reviewer #2: (No Response)

Reviewer #3: Overall this is a wellexecuted and highly relevant study with solid methodological rigor and clear public health implications. The manuscript is suitable for publication with only few minor revisions and editorial improvements to enhance clarity and readability

1-writing and language:

Some sections particularly in the Methods and Results are dense with technical detail. Consider simplifying or breaking up long paragraphs for better flow.

2- figures and tables:

Table 1 is comprehensive but quite lengthy. Consider splitting it into two parts (e.g., individual and household level variables) or moving some less critical rows to supplementary material to enhance focus

3-terminology:

Ensure consistent terminology across the manuscript for WASH indicators (eg., unimproved toilet and unimproved toilet facilities) and geographic regions (eg., North Yemen and northern Yemen).

Reviewer #4: Moderate Revision

**Summary and General Comments**

Reviewer #1: No comments

Reviewer #2: (No Response)

Reviewer #3: This manuscript gives a comprehensive and timely analysis of diarrhea prevalence among children under five in Yemen over a 30 year period, utilizing nationally representative DHS and MICS surveys. The authors apply multi level logistic regression models to explore both individual and household level factors that influencing diarrhea risk, while also comparing Yemen’s situation to global trends. The study is highly relevant given the ongoing humanitarian crisis in Yemen and offers important insights into public health disparities which is reminded me of what’s happening in sudan and south sudan because of the ongoing conflicts

2. Strengths

1-Robust dataset: The useing of four large scale, population based surveys across multiple decades provides solid ground for temporal analysis.

2-Methodological rigor: The application of three level random effects logistic regression is appropriate and adds depth, especially in capturing clustering effects.

3-Global context: The inclusion of global prevalence comparisons through DHS metaanalysis enhances the broader relevance of the findings.

4-Policy relevance: The findings are actionable and offer clear implications for improving WASH infrastructure and targeting vulnerable populations

A few minor weaknesses to consider:

1-The text, while thorough, can be dense in places. Streamlining certain sections and reducing redundancy would improve flow.

2-While ethical approval is appropriately addressed, a brief note on how missing data were handled in practical terms (e.g., any imputation and listwise deletion) could add clarity.

In terms of novelety, this study fills a major gap in the literature by moving beyond small, hospital based studies to provide a national picture with longitudinal insights. The findings are significant for public health planning in Yemen and other conflict affected countries such as Sudan , South sudan , Syria and Palestine where access to water, sanitation, and healthcare is compromised.

There’s no concerns regarding publication ethics

Reviewer #4: This manuscript presents a well-conceived and methodologically rigorous study addressing the burden of childhood diarrhea in Yemen. The use of nationally representative data and advanced three-level random-effects logistic regression modeling strengthens the analysis. The findings are highly relevant for public health policy, particularly in settings affected by humanitarian crises. However, the manuscript would benefit from clarifications in statistical interpretation, stronger policy recommendations, and improved contextual analysis of ongoing health system interventions and external factors affecting disease prevalence.

The introduction contains excessive background detail—consider streamlining for better readability.

The study does not discuss vaccination programs, particularly rotavirus vaccine introduction in Yemen—a key preventive measure against diarrheal diseases. If available, information on coverage rates, effectiveness, and impact should be included to strengthen the disease prevention aspect of the analysis.

Yemen has undergone significant population displacement due to conflict, economic instability, and natural disasters. How has internal migration, refugee influx, or rural-to-urban shifts affected diarrhea prevalence and access to healthcare? Integrating mobility trends could provide a deeper understanding of geographic disparities in disease burden.

PLOS authors have the option to publish the peer review history of their article (what does this mean? ). If published, this will include your full peer review and any attached files.

**Do you want your identity to be public for this peer review?** For information about this choice, including consent withdrawal, please see our Privacy Policy .

Reviewer #1: **Yes: ** Mohamed Ali Daw, MD, MPS, PhD, MRCPath (1), FTCDI

Reviewer #2: No

Reviewer #3: No

Reviewer #4: **Yes: ** Dr. Fekri Dureab

---

## [Decision Letter · Decision Letter 1]

Dear Mr. Tag,

We are pleased to inform you that your manuscript 'Diarrhea in Yemeni Children Under Five: A Multi-Level Analysis of Population-Based Surveys, 1991-2022' has been provisionally accepted for publication in PLOS Neglected Tropical Diseases.

Best regards,

Qu Cheng, Ph.D.

Section Editor

Qu Cheng

Section Editor

Shaden Kamhawi

co-Editor-in-Chief

Paul Brindley

co-Editor-in-Chief

Reviewer's Responses to Questions

**Key Review Criteria Required for Acceptance?**

**Methods**

-Are the objectives of the study clearly articulated with a clear testable hypothesis stated?

-Is the study design appropriate to address the stated objectives?

-Is the population clearly described and appropriate for the hypothesis being tested?

-Is the sample size sufficient to ensure adequate power to address the hypothesis being tested?

-Were correct statistical analysis used to support conclusions?

-Are there concerns about ethical or regulatory requirements being met?

Reviewer #2: Yes

**Results**

-Does the analysis presented match the analysis plan?

-Are the results clearly and completely presented?

-Are the figures (Tables, Images) of sufficient quality for clarity?

Reviewer #2: Yes

**Conclusions**

-Are the conclusions supported by the data presented?

-Are the limitations of analysis clearly described?

-Do the authors discuss how these data can be helpful to advance our understanding of the topic under study?

-Is public health relevance addressed?

Reviewer #2: Yes

**Editorial and Data Presentation Modifications?**

Reviewer #2: (No Response)

**Summary and General Comments**

Reviewer #2: (No Response)

PLOS authors have the option to publish the peer review history of their article (what does this mean? ). If published, this will include your full peer review and any attached files.

**Do you want your identity to be public for this peer review?** For information about this choice, including consent withdrawal, please see our Privacy Policy .

Reviewer #2: No

---

## [Editor Report · Acceptance letter]

Dear Mr. Tag,

We are delighted to inform you that your manuscript, "Diarrhea in Yemeni Children Under Five: A Multi-Level Analysis of Population-Based Surveys, 1991-2022," has been formally accepted for publication in PLOS Neglected Tropical Diseases.

Best regards,

Shaden Kamhawi

co-Editor-in-Chief

Paul Brindley

co-Editor-in-Chief
